# Hot Topics on Fertility Preservation for Women and Girls—Current Research, Knowledge Gaps, and Future Possibilities

**DOI:** 10.3390/jcm10081650

**Published:** 2021-04-13

**Authors:** Kenny A. Rodriguez-Wallberg, Xia Hao, Anna Marklund, Gry Johansen, Birgit Borgström, Frida E. Lundberg

**Affiliations:** 1Department of Oncology and Pathology, Karolinska Institutet, SE-171 64 Stockholm, Sweden; xia.hao@ki.se (X.H.); anna.marklund@sll.se (A.M.); gry.johansen@ki.se (G.J.); birgit.borgstrom@ki.se (B.B.); frida.lundberg@ki.se (F.E.L.); 2Department of Reproductive Medicine, Division of Gynecology and Reproduction, Karolinska University Hospital, SE-141 86 Stockholm, Sweden

**Keywords:** fertility preservation, cancer, chemotherapy, obstetric outcome, reproductive epidemiology, genetic conditions, infertility, cryopreservation, assisted reproductive technology, premature gonadal failure

## Abstract

Fertility preservation is a novel clinical discipline aiming to protect the fertility potential of young adults and children at risk of infertility. The field is evolving quickly, enriched by advances in assisted reproductive technologies and cryopreservation methods, in addition to surgical developments. The best-characterized target group for fertility preservation is the patient population diagnosed with cancer at a young age since the bulk of the data indicates that the gonadotoxicity inherent to most cancer treatments induces iatrogenic infertility. Since improvements in cancer therapy have resulted in increasing numbers of long-term survivors, survivorship issues and the negative impact of infertility on the quality of life have come to the front line. These facts are reflected in an increasing number of scientific publications referring to clinical medicine and research in the field of fertility preservation. Cryopreservation of gametes, embryos, and gonadal tissue has achieved quality standards for clinical use, with the retrieval of gonadal tissue for cryopreservation being currently the only method feasible in prepubertal children. Additionally, the indications for fertility preservation beyond cancer are also increasing since a number of benign diseases and chronic conditions either require gonadotoxic treatments or are associated with premature follicle depletion. There are many remaining challenges, and current research encompasses clinical health care and caring sciences, ethics, societal, epidemiological, experimental studies, etc.

## 1. Introduction

The preservation of fertility potential is an important issue for young people at risk of becoming infertile due to clinical conditions, diseases, or treatments. For patients with cancer, the current international guidelines recommend timely information on the risk of fertility loss associated with planned treatment, and referral to appropriate reproductive medicine centers to discuss further options for fertility preservation [1,2,3]. Cancer is a heterogeneous disease; Figure 1 illustrates the various approaches for cancer treatment and the options for fertility preservation that can be considered in the respective cases. Although several methods of fertility preservation are clinically available, studies have shown that young patients at risk of infertility are not always aware of these options. Some patients are not offered fertility counseling at all due to the lack of appropriate information and education of the medical teams, while other patients are not able to recall the information that is provided at a time of emotional distress [4,5,6,7,8]. 

Fertility preservation is also indicated when treatment of benign conditions may encompass a risk of infertility, either through medical treatments or surgery [9,10]. Some genetic conditions, such as the Turner syndrome, are associated with a shorter reproductive span and predisposition to premature gonadal insufficiency in females, and consequently, these have also become indications for fertility counseling and fertility preservation. The list of indications is growing, ranging from common gynecologic diseases such as endometriosis to chronic autoimmune diseases or rare genetic syndromes, and the number of patients who undergo procedures for fertility preservation is continuously increasing. Some of the most common diagnoses indicative of procedures for fertility preservation in women and girls and their current challenges are outlined below. Research in the field is well supplied, highlighting the need to investigate clinical outcomes by follow up of all individuals undergoing interventions for fertility preservation to determine the safety of the procedures in the long term and their efficacy. Experimental research is also needed since several of the methods are still under development [2,3,4]. 

## 2. Fertility Preservation in Children

Cryopreservation of gonadal tissue aiming at its potential use in future reproductive treatments has been done successfully for many years in adult patients. In the case of pre-pubertal children, this method is their current only option to cryopreserve reproductive potential [11,12]. In adolescents, additional methods to cryopreserve mature gametes can be practiced, similarly to as in adults, if they have completed puberty development [13]. As evidenced in an increasing number of publications, fertility preservation in prepubertal children is a specifically demanding area for research due to the naturally immature status of the gonads. The situation is similar for children in early puberty, at Tanner stages 2–3 [14]. To retrieve the ovarian tissue for cryopreservation, the performance of a unilateral oophorectomy or retrieval of small ovarian biopsies via laparoscopy has been reported by several centers [11,15,16]. It is important to highlight that the methods chosen for fertility preservation should not cause any delay in starting planned cancer treatment and that surgery for gonadal tissue retrieval in children should only be performed in absence of low blood counts, thrombocytopenia, or any other vulnerabilities posing an increased surgery risk for the child [16,17]. The preferred time point for retrieval of gonadal tissue for cryopreservation aiming at fertility preservation in children is obviously before the initiation of any chemotherapy rounds. However, this is not always possible, and research indicates that there are several feasible time points regarding these young patients. Figure 2 indicates possible time points (gray boxes A–D) proposed for fertility preservation of children presenting with acute lymphocytic leukemia (ALL) and undergoing treatments, including hematological stem cell transplantation, as an example. The proportions of individuals at each of the figure’s boxes are approximately calculated, according to current data from the Nordic Society for Pediatric Hematology and Oncology (NOPHO) regarding the ALL 2008 Protocol for Childhood Acute Lymphoblastic Leukemia. Predicted overall survival at follow up in these patients is currently about 93% [17]. 

The retrieval of gonadal tissue by invasive methods in prepubertal children is not a clinical standard practice for several reasons. An important issue is a risk for malignant cells in the tissue, often seen in children with leukemia, one of the most frequent pediatric malignancies [17]. If re-transplantation of preserved tissue containing malignant cells is performed, there is a risk that these cells could cause a relapse. This risk could be eliminated with the use of in vitro methods to obtain mature germ cells [18]. Another reason is the complicated question of whether the gonad biopsy in itself could diminish the chance of developing natural fertility in adult age. There is a wide variation in opinions on how high the risk for future infertility should be in order to perform an invasive biopsy in a child [17]. Additional difficulties include the fact that current treatments for malignant diseases are still evolving, and a precise calculation of a future infertility risk is difficult, considering that the desire to create a family and face fertility issues will be realized by 10–20 years into the future. Thus, invasive procedures in children for gonadal retrieval under these conditions should be performed under ethical review board-approved studies [19].

On the other hand, girls after menarche could benefit from the same fertility preservation methods that are applicable to adults. However, difficulties arise in counseling young individuals that may not expect that fertility preservation may be of value for their future. The fact that counseling on fertility preservation after confirmation of a cancer diagnosis is usually managed as an emergency also makes communication difficult and lead to situations in which fertility preservation is finally not performed, even when it would have been possible and beneficial to the young patient at risk of infertility [20].

There are several research areas that need to be developed to solve the technical problems with fertility preservation in prepubertal children. For teenagers, we need a better psychological understanding of how to help these young people make a decision that they will not regret in the future [19].

## 3. Fertility Preservation in Girls and Teenagers Presenting with Turner Syndrome

One of the most characteristic symptoms of Turner syndrome, and the most distressing to the women with the syndrome, is infertility due to ovarian failure and the typical streak ovaries. There are, however, girls with Turner syndrome who present with signs of ovarian function in childhood and adolescence but have a risk of development of premature ovarian failure. Approximately 20% of girls with Turner syndrome develop some pubertal symptoms, usually breast development, but only 2–3% proceed to menarche [21]. It is currently recommended to offer counseling on reproductive issues when a Turner diagnosis is confirmed and also perform a careful investigation of cardiovascular features that may contraindicate a future pregnancy [22].

The recommended way to find the girls that have functioning ovaries and provide advice on fertility preservation at an early age is by taking repeated anti-Müllerian hormone (AMH) tests during childhood and adolescence [23,24]. AMH is a routine test in many hospital laboratories, so it is likely available in most clinics taking care of girls with Turner. The chance to have ovarian function is better in mosaicism (45X/46XX, 45X/47XXX) than in the other chromosome variants (46X with ring chromosome, 46X with deletion, 46X with isochromosome, or 45X) [25]. However, there are exceptions, and while the Turner diagnosis is usually based on a chromosomal analysis from blood tests, it is possible that the chromosome variants in ovarian tissue differ from blood. 

If AMH is detected it should be followed regularly, i.e., once a year or when other blood tests are needed, for example, those required under growth hormone treatment, hypothyroidism, or celiac disease, which are common in the management of girls with Turner syndrome.

When the girl is mature enough to understand a discussion about her future fertility, she may be offered an ovarian biopsy via a minimally invasive laparoscopy [26]. Information about chances and limitations in the future is important. Re-transplantation of ovarian tissue in adult age is a possibility in Turner, although there have been no reports on successful fertility treatment using prepubertal ovarian tissue so far. However, the risks seen in children with malignant disease are not an issue for this patient group.

If the girl reaches spontaneous menarche, ovarian stimulation and collection of oocytes could be considered [27,28]. After a successful procedure resulting in a number of mature oocytes cryopreserved, the chances for fertility treatment resulting in children in the future is good [16]. Considering the much better prognosis of this procedure, ovarian stimulation can also be offered to girls who have previously cryopreserved ovarian tissue. In all discussions of fertility preservation in women with Turner syndrome, other medical problems that may limit the chances to fulfill a pregnancy using own-preserved or donated oocytes need to be considered [22,26,29]. In general, cardiac examinations should be conducted and regularly repeated during follow up in women with Turner syndrome to determine if there is any significant aorta dilation in relation to the body surface. Some patients may require blood pressure lowering and/or surgery. Congenital aberrations (bicuspid aortic valve, coarctation of the aorta) and/or aortic dilatation are strongly associated with life-threatening aortic dissection. In line with international guidelines, medical assistance to achieve pregnancy is not recommended if the congenital heart and/or vessel abnormalities (surgically corrected or not), or acquired aortic dilation is present in women with Turner syndrome [22]. Other options for parenthood should be advised.

## 4. Fertility Preservation in Women with Breast Cancer

Breast cancer is the most common malignancy affecting women of reproductive age [30]. Advances in screening and treatment methods have significantly improved survival rates for patients with breast cancer, with predicted five-year relative survival close to 90% [31]. Young age at diagnosis is associated with increased risk for aggressive biological features and advanced stage of disease at presentation [32,33]. Therefore, (neo) adjuvant chemotherapy and adjuvant endocrine therapy are frequently recommended for these patients [34]. Adverse effects of chemotherapy include gonadal damage, while several-years-long endocrine treatment in women with estrogen receptor-positive breast cancer results in developing age-related infertility. Fertility counseling, including information on existing methods of fertility preservation, should be offered prior to the start of antineoplastic treatments to all women diagnosed with breast cancer at reproductive age [2]. In large programs for fertility preservation, breast cancer is consistently the most common diagnosis, thus allowing to perform studies of large sample size in this patient group.

For adult women, established methods of fertility preservation include cryopreservation of oocytes, embryos, and ovarian tissue [35,36,37,38]. The banking of oocytes and embryos are the most commonly used methods in women with breast cancer. Both these methods require controlled ovarian stimulation with gonadotropins (COS), which usually can be completed during a couple of weeks. The process is followed by follicular aspiration, usually transvaginally and ultrasound guided to obtain the oocytes. The production of embryos requires additional methods for in vitro fertilization (IVF) available at IVF-laboratories worldwide. The time window between breast cancer surgery and the start of chemotherapy usually provides enough time to complete this procedure. COS results in a short-term increase of systemic estradiol levels [39]. Since estrogen is a growth factor for breast tissue, a potentially safer approach to COS has been developed for patients with estrogen-sensitive diseases—a protocol using aromatase inhibitor letrozole [40,41]. Another recent improvement is the use of gonadotropin-releasing hormone analogs (GnRHa) instead of human chorionic gonadotropin for ovulation trigger, which further reduces estradiol levels after oocyte pickup and minimizes the risk of ovarian hyperstimulation [42]. Random start, meaning the immediate start of COS irrespective of the phase of the menstrual cycle, allows the interval from referral to oocyte retrieval to be considerably shortened and usually kept within two weeks [43,44,45]. Although the main aim of these approaches is to increase the safety of COS in women with breast cancer, they have also been reported to result in similar reproductive outcomes, measured as numbers of oocytes and embryos cryopreserved [46]. 

Overall, the safety of fertility preservation in the setting of breast cancer, including the safety of COS, has been investigated in several recent studies [45,47,48,49,50]. No deterioration in prognosis for relapse-free or overall survival has been observed so far, but follow-up time has been relatively short and the number of included patients relatively small. Moreover, several studies report overall survival as a proxy for the safety of COS in women with breast cancer, while relapse rate in women with and without fertility preservation would be a more adequate outcome for investigation of safety in this population. 

Long-term efficacy of fertility preservation in populations of young women with breast cancer can be measured as utilization rates of cryopreserved specimens after completed treatment or as subsequent pregnancy and live-birth rates. These types of data are yet scarce. Pregnancy rates after the use of cryopreserved embryos in survivors of breast cancer were reported to be comparable to those expected after in vitro fertilization in the general infertile population [51]. In a cohort of 118 women counseled on fertility preservation at the time of breast cancer diagnosis, a five-year live-birth rate of 29.4% was reported in women with fertility preservation, compared to 19% in women without fertility preservation [52]. In a Swedish nationwide cohort study including 1275 women with breast cancer, fertility preservation at the time of diagnosis was associated with a 2.3 times higher likelihood of giving birth to a child after completed treatment of breast cancer, with a five-year cumulative incidence of postcancer live births of 19.4% among women exposed to fertility preservation and 8.6% among comparators [50]. An important limitation in interpreting these results is that adjustment for childbearing wishes at the time of diagnosis could not be performed. 

The stage of maturity of the oocytes obtained is important since only mature oocytes at Metaphase II (MII) can be fertilized. If the oocytes will be cryopreserved, the method currently used is by vitrification, and immature oocytes are discarded [35,36]. However, the aim to obtain mature MII oocytes following COS is not always accomplished, and studies have been performed to evaluate the ability to mature the oocytes that are retrieved at immature stages (Germinal vesicle, GV, of Metaphase I, MI) at the IVF laboratory. The process is known as in vitro maturation (IVM), and it has been reported to enhance the yields of oocytes and embryos in women with breast cancer undergoing COS for fertility preservation [53]. A recent innovative procedure has also been reported in a woman with breast cancer that did not undergo COS, and the follicle aspiration was performed in the natural cycle. The oocytes were thus obtained immature and underwent IVM and vitrification. Thereafter, the oocytes were thawed and fertilized resulting in embryo transfer and a live birth [54]. In general, methods for IVM and in vitro culture of follicles obtained from ovarian tissue are highly needed [18] to enhance the options of fertility preservation for female patients.

## 5. Fertility Preservation in Women with Gynecological Cancer

Long-term survival rates after gynecological cancer, especially in young patients, are increasing, and surgical techniques aimed at sparing reproductive organs and preserving fertility have been developed. The term most used for this type of surgery is fertility-sparing surgery (FSS), and it is often possible in early gynecological cancers [55]. Although research has been extensive in the field, only a few prospective studies have been conducted to date. 

Ovarian cancer is in general a disease appearing at an older age, and only about 8% of the women diagnosed with ovarian cancer are under 40 years of age [56]. Ovarian cancer encompasses a broad spectrum of diseases, divided into epithelial ovarian cancer, including borderline ovarian tumors, and non-epithelial ovarian cancer, which includes sex cord-stromal cell tumors and malignant ovarian germ cell tumors.

The primary treatment of ovarian tumors is extensive surgery including hysterectomy and bilateral salpingo–oophorectomy, which results in infertility. Depending on the histologic subtype, surgery is combined with adjuvant chemotherapy. An FSS encompasses the practice of sparing the uterus and at least a part of one ovary, and it can be considered in women of childbearing age who wish to preserve their fertility, but this is also depending on the tumor type and the stage at diagnosis.

In young women with early-stage non-epithelial ovarian cancer, FSS is now considered the standard surgical treatment [57,58,59], and it can be considered even in the advanced stage of malignant ovarian germ cell tumors, due to the high chemosensitivity of these tumors [57]. Fertility is reported to be maintained after treatment with adjuvant chemotherapy for this type of tumor and the majority of women present with regular menstrual cycles during follow up [60]. 

Borderline ovarian tumors (BOTs) are tumors with epithelial origin but without stromal invasion. Compared to malignant ovarian tumors, BOTs have better prognoses, and FSS is an accepted treatment for women of reproductive age [61,62]. Even though the recurrence rate in BOTs is reported to be higher after FSS, compared with women undergoing radical surgery, it does not affect the overall survival [61]. The risk of malignant transformation of BOTs must be considered in all cases, and the women should be advised to undergo radical surgery after they have completed their desired family size [61,63]. While FSS is an accepted treatment in both early and advanced stage non-epithelial ovarian cancer and BOTs, there is no current international consensus about the stage or grade at which FSS should be considered as safe as radical surgery in epithelial ovarian cancers with specific histological subtypes of tumors of high malignancy potential [58,64]. FSS is an accepted treatment for stage IA and grade 1 tumors, while stage IC and grade 3 tumors and clear cell histology are considered risk factors for poorer prognosis [65]. Isolated ovarian recurrence, which generally has a good prognosis, is most often seen in tumors with good prognostic factors, while extra ovarian recurrence to a greater extent is seen in stage IC and grade 3 tumors [66,67]. There is an ongoing debate on whether radical surgery will improve the prognosis of those tumors [66,68]. 

Cervical cancer, which in most cases is caused by infection with high-risk oncogenic human papillomavirus (HPV), is commonly diagnosed in young women, and about one-third of all women who are diagnosed with cervical cancer in Sweden each year are under 40 years of age [56]. In the early stages of the disease, the standard treatment is radical hysterectomy with pelvic lymphadenectomy, and more advanced stages are treated with radiation and combined chemotherapy [56,69]. Preservation of fertility can only be considered in the early stage of the disease if the tumor is <2 cm and of nonaggressive histological subtypes [69,70]. Depending on the oncologic characteristics of the tumor, there are different surgical techniques that are available as options for fertility preservation, such as cone resection, vaginal trachelectomy, laparotomic, laparoscopic, and robot-assisted abdominal radical trachelectomy. If the tumor exceeds 2 cm, trachelectomy combined with neoadjuvant chemotherapy can be considered in selected cases of squamous cell carcinoma where >10 mm of the cervix is tumor-free, which should be evidenced on MRT and advanced ultrasound examination [56]. 

In endometrial cancer, which is the most common gynecological cancer overall, only around 2% are diagnosed in women below 40 years of age [56]. In the early stage of the disease with complex atypical endometrial hyperplasia and grade I endometrial carcinoma, a conservative approach including hormonal therapy, hysteroscopic resection of focal lesions, and intrauterine progestin delivery devices, can be considered as an option for fertility preservation in women of reproductive age [71].

## 6. Epidemiological Research of Long-Term Outcomes of Fertility Preservation

Since randomized clinical trials of fertility preservation in women at risk of infertility vs. no intervention may be unethical to perform, observational epidemiological studies following these two groups are key to evaluate the safety and efficacy of fertility preservation. The results summarized in the previous sections of this review indicate that ovarian stimulation for fertility preservation is safe for breast cancer patients and selected patients undergoing fertility-sparing surgery for the treatment of gynecological cancer. However, for other cancer types, the data available are limited. In a US study from 2018, no statistically significant difference in cancer recurrence or mortality after ovarian stimulation for fertility preservation in patients with breast (*n* = 262), hematological (*n* = 95), gynecological (*n* = 65) or other cancer (*n* = 75) were reported after a median follow up of 3.8 years [72]. Another US study on 128 women with lymphoma reported a longer time to treatment initiation in patients receiving ovarian stimulation for fertility preservation but found no statistically significant difference in progression-free survival after five years [73]. 

For the investigation of the efficacy of fertility preservation in female cancer survivors, data on the women who return to use cryopreserved specimens from fertility preservation are scarce. The estimated live-birth rates using cryopreserved embryos or oocytes range from 22–45% per embryo transfer in studies published so far [51,74,75,76,77]. A large Spanish multicenter study including 1073 cancer patients who had cryopreserved oocytes reported on 80 patients that returned to attempt pregnancy with a cumulative live-birth rate of 35% [78]. Some of the women who undergo fertility preservation may also be able to conceive without the use of cryopreserved specimens, i.e., by natural conception, or undergoing new treatment using assisted reproductive technology with autologous oocytes or by using donor oocytes. Although cryopreservation of ovarian tissue has been considered an experimental method of fertility preservation until recently, more than 200 live births following re-transplantation have been reported with live-birth rates of 25–30% [79,80,81]. In addition, new experimental methods are being developed, such as the recently reported cases of women that achieved live birth following embryo transfer using oocytes recovered from ovarian tissue and matured in vitro [82]. Regarding fertility-sparing surgery, the overall pregnancy rate is estimated at 42%, according to a systematic review that included more than 11,000 patients with gynecological cancer [83]. 

It is evident that larger cohorts with longer follow up are needed to confirm the long-term safety of fertility preservation, especially for less common cancers. Detailed information on tumor characteristics and other prognostic factors is also needed to ensure comparability between cancer patients with and without fertility preservation. In order to evaluate the efficacy of fertility preservation, future studies should ideally include a comparison group who did not undergo fertility preservation, in addition to data on childbearing intent at the time of diagnosis and pregnancy attempts after cancer for all patients, which are lacking in the current publications. To investigate the long-term reproductive outcomes of fertility preservation treatments further, detailed data on pregnancies and live births such as mode of conception, obstetric, and neonatal outcomes are needed. Pooling data from countries with similar registers and health care structures may be an option to provide sufficient power for these comparisons, although access to fertility preservation has been shown to vary greatly between countries [84].

The establishment of national and international registries of fertility preservation will further improve the possibilities to follow this patient population. The European Society of Human Reproduction and Embryology (ESHRE) recently started collecting data on fertility preservation in an optional module of the ESHRE IVF-monitoring scheme [36]. According to the latest report, twelve European countries were able to provide data on fertility preservation treatments performed in 2015 [85]. In Australia and New Zealand, the Fertility Understand Through Registry and Evaluation (FUTuRE) team has established the Australasian Oncofertility Registry with the aim to collect fertility preservation data from cancer and fertility centers across the two countries [86]. These and other similar initiatives are likely to provide more comprehensive data on fertility preservation treatments in larger cohorts of patients in the future.

## 7. Pharmaceutical Research and Stem Cell Strategies in Fertility Preservation

There are numerous experimental studies currently ongoing aiming at developing possible pharmaceutical fertility preservative substances, which can be administrated prior to or during ovarian-toxic treatment to prevent its damage to ovaries [87]. Published studies in this field have been mainly designed according to the proposed mechanisms of how the ovaries are damaged by gonadotoxic treatment, especially chemotherapy. The explanation of chemotherapy-induced infertility has been mainly attributed to inducing primordial follicle depletion that results in diminished ovarian reserve. The involved mechanisms proposed include induction of primordial follicle apoptosis triggered by DNA damage and/or oxidative stress [88,89,90,91,92,93,94,95]. Additional proposed mechanisms to explain primordial follicle depletion include the overactivation effect of the gonadotoxic chemotherapy on the dormant primordial follicles through the activation of the PI3K/PTEN/Akt signaling pathway, which results in a premature “burnout” of the ovarian reserve [87,96,97,98,99,100]. Chemotherapy has also been suggested to damage the ovarian blood vessels [101,102,103]. Although all described mechanisms are plausible, a very recent study demonstrated cyclophosphamide-induced primordial follicle depletion in a human ovarian xenograft model by means of triggering proapoptotic pathways, without evidence of primordial follicle activation, and indicated apoptosis as the main mechanism [104]. 

Antioxidants or substances with antioxidative effects, bilberry, mirtazapine and hesperidin, mesna, sildenafil citrate, and hydrogen-rich saline have been shown to be able to reduce ovarian damage induced by cisplatin or cyclophosphamide in rats [105,106,107,108,109]. Dexrazoxane may attenuate ovarian damage induced by doxorubicin by preventing DNA damage and the activation of gamma-H2A histone family member X in mouse and marmoset ovarian tissues [110,111]. Imatinib has been shown to specifically protect mouse ovaries from cisplatin damage by preventing apoptosis in oocytes and preserving the follicle reserve [88]. An inhibitor of the protein kinase ataxia–telangiectasia mutated (ATM) that regulates response to DNA damage, KU55933, seems to protect rat ovaries from cyclophosphamide damage by inhibiting apoptosis induced by ATM activation [89]. Other than ATM, ataxia–telangiectasia and Rad3-related protein (ATR) is another essential regulator of DNA damage checkpoints [112]. Inhibition of ATR or checkpoint kinase 2 (Chk2, an effector of ATM) could protect mouse ovaries from cyclophosphamide-induced primordial follicle loss [94]. Sphingosine-1-phosphate, an inhibitor of ceramide-promoted apoptosis, has been shown to reduce busulfan-induced ovarian follicle apoptosis in mice and cyclophosphamide or doxorubicin-induced ovarian follicle apoptosis in human ovarian tissue xenografts [113,114]. AMH co-treatment with chemotherapy has also shown significant preservation of primordial follicles in mice by inhibiting their activation [115]. Ammonium–trichloro (0,0-dioxyethylene) tellurate could also inhibit the activation of primordial follicles following cyclophosphamide treatment in mice [97]. Rapamycin, an inhibitor of the mammalian target of rapamycin complex 1, could also prevent cyclophosphamide-induced primordial follicle activation [100,116]. Melatonin has shown a similar effect against cisplatin through mediating phosphatase and tensin homolog and inhibiting protein kinase B, glycogen synthase kinase 3 beta, and forkhead box O3 activation in mice [117]. Granulocyte colony-stimulating factor co-treatment during cyclophosphamide and busulfan could increase microvessel density and improve the fertility competence of mice after chemotherapy [118]. 

As regards human experimental studies, the fertility preservative effect of Gonadotropin-releasing hormone analogs (GnRHa) during chemotherapy has been investigated in several clinical trials. The data are still controversial. The guidelines for female fertility preservation of American Society of Clinical Oncology (ASCO) 2018, the ESHRE 2020, and the European Society for Medical Oncology (ESMO) 2020 do not recommend GnRHa as a proven fertility preservative method and stated that it should not thus be used in that place [2,36,37]. Although a few studies suggest higher numbers of women that achieved pregnancies in the GnRHa-treated groups [119], all studies have been unblinded; thus, the bias of higher pregnancy attempts in the women that have received the GnRHa treatment in randomized studies has been discussed [120,121,122]. 

Additionally, there are several studies aiming to restore or rescue fertility competence after alkylating drug treatment using stem cells with mesenchymal origin. These stem cells could be isolated either from rodent femurs, tibiae, or adipose tissue or from human circulating blood, menstrual blood, amniotic fluid, placenta, or chorionic plate of the placenta. When different types of stem cells were injected into tail veins or ovaries of cyclophosphamide and/or busulfan or cisplatin sterilized/treated rodents, all of them showed improved fertility competence [123,124,125,126,127,128,129,130,131,132,133,134,135,136]. Of special note, there is one study on human subjects, which injected autologous iliac crest-derived bone marrow mesenchymal stem cells into the ovaries of ten women with premature ovarian insufficiency, in which two women recovered menstruation and one got pregnant [137]. 

## Figures and Tables

**Figure 1 jcm-10-01650-f001:**
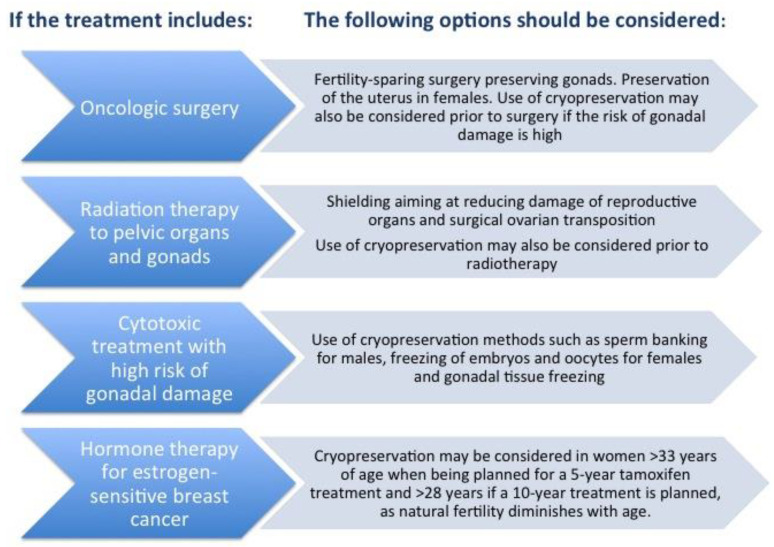
Fertility preservation strategies depending on the type of oncological treatment in females and males. Reprinted with permission from Rodriguez-Wallberg and Oktay. *Cancer Management and Research* 2014, 6, 105–117; originally published by and used with permission from Dove Medical Press Limited.

**Figure 2 jcm-10-01650-f002:**
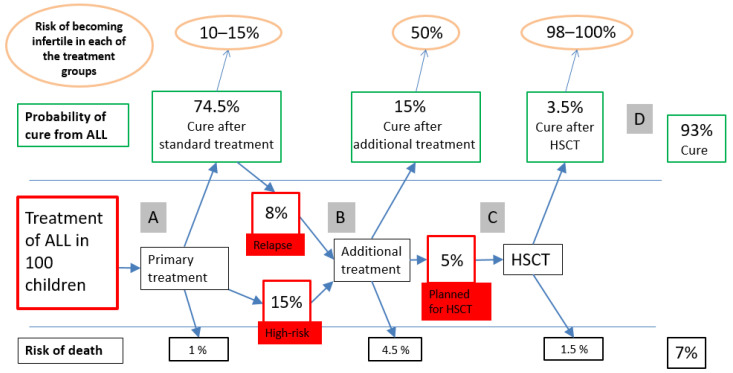
Possible time points (A–D) for fertility preservation of children presenting with acute lymphocytic leukemia (ALL). Reprinted with permission from Borgström et al. *Pediatr Blood Cancer*. 2020, 67, e28507. HSTC = hematological stem cell transplantation [17].

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
