# Peer review of "Hot Topics on Fertility Preservation for Women and Girls—Current Research, Knowledge Gaps, and Future Possibilities"

_jcm, 2021, doi:10.3390/jcm10081650_

Round 1
Reviewer 1 Report
Fertility preservation is one of the most important aspect in the contemporary gynecology (more precisely reproductive endocrinology).
Presented manuscript focuses on all important problems of this topic. Authors described experimental research in fertility preservation. In my opinion it is particularly essential. Additionally field of fertility preservation in children was presented. Special attention was paid to fertility preservation in Turner syndrome patients. Finally aspects of fertility preservation in women with breast cancer and gynecological cancers were described.
In my opinion it is very well prepared review on contemporary status of fertility preservation in girls and women
Author Response
We thank the reviewer for the positive and encouraging comments on our manuscript.
We have revised the language, according to your recommendation for minor spell check.
Reviewer 2 Report
Rodriguez-Wallberg et al present an editorial on Hot topics on fertility preservation for women and girls – Current research, knowledge gaps and future possibilities. I found their work interesting, however I have some concerns:
Figure 1 is very generics. It gives some impact to the article, however I would suggest to make a figure either differentiating further to the procedures that can be implemented for young girls vs adults or only to make a figure presenting the “hot topics” and new techniques
Paragraph 2: there is information only on very preclinical studies and little on the more advanced ones (ex mixing of techniques), freezing techniques etc
Paragraph 3: I would add information on eventual hormonal stimulation (and its possible duration) before oocyte preservation
Lines 156-158 I would suggest to comment on the differences between hematological and non haematological malignancies rather than ALL only
I would suggest to make clear divisions between options that can be used in fertility preservations of girls (pre/post pubertal), AYA and older women
I would suggest to raise the topic of clinical staging of patient upon diagnosis and the amount of time one has before the start of oncological treatment.
I would also suggest to describe which procedures carry risks associated with surgical procedures and which can be used according to the patients wellbeing etc.
I find the paragraph on the management of female with Turner syndrome very attractive.
Author Response
Rodriguez-Wallberg et al present an editorial on Hot topics on fertility preservation for women and girls – Current research, knowledge gaps and future possibilities. I found their work interesting, however I have some concerns:
Figure 1 is very generics. It gives some impact to the article, however I would suggest to make a figure either differentiating further to the procedures that can be implemented for young girls vs adults or only to make a figure presenting the “hot topics” and new techniques
Response: Thank you for your comments. Please see our responses point by point here below. The texts and changes have been highlighted in yellow in the revised manuscript.
We wish to maintain the figure 1, as it illustrates how heterogenous are the treatments for cancer in women and also the current available options for fertility preservation that have been developed. In the chapter of children the issues of cryopreservation of gonadal tissue are described in detail. In that context the figure 1 is illustrative for our introductory section.
Paragraph 2: there is information only on very preclinical studies and little on the more advanced ones (ex mixing of techniques), freezing techniques etc
Response: Discussion of cryopreservation methods was outside of the scope of our review, as the methods have been established and proved efficacious. We were interested in discussing the specific challenges of several patient groups as hot topics, and the rapidly moving research on pharmacological experimental research.
Paragraph 3: I would add information on eventual hormonal stimulation (and its possible duration) before oocyte preservation
Response: The duration of hormonal stimulation and the improvements in stimulation recently developed, such as random-start, agonist trigger or addition or letrozole are all described in detail in the chapter on breast cancer. The text has been highlighted, second paragraph of the chapter on breast cancer.
Lines 156-158 I would suggest to comment on the differences between hematological and non haematological malignancies rather than ALL only
Response: The use of the ALL data is only illustrative of the several timepoints that can be applicable for fertility preservation of children, inclusive those being treated with stem cells transplantation.
I would suggest to make clear divisions between options that can be used in fertility preservations of girls (pre/post pubertal), AYA and older women
Response: Thank you, we have added these details in the first paragraph of chapter 2.
I would suggest to raise the topic of clinical staging of patient upon diagnosis and the amount of time one has before the start of oncological treatment.
I would also suggest to describe which procedures carry risks associated with surgical procedures and which can be used according to the patients wellbeing etc.
Response: Thank you for your comment. We have clarified these aspects in the chapter of children, and added a sentence, as fertility preservation should not represent an increased surgical risk for the patient and should not cause any delay in a planned cancer treatment.
I find the paragraph on the management of female with Turner syndrome very attractive.
Response: Thank you for your comments.
Reviewer 3 Report
This is a well written review about fertility preservation for women and girls.
I have just some concerns listed below:
- In vitro maturation (IVM) technique is not mentioned in this review although it “ should be regarded as an innovative FP procedure “ as recommended by ESHRE fertility preservation guideline published in 2020. Even if oocyte or embryo cryopreservation after IVM is less effective than after COS, it can proposed in specific indication or when letrozole cannot be proposed.
I think that this technique should be at least mentioned in this review ( for example in the breast cancer part as a live birth was recently described Grynberg et al, 2020)
- Authors decided to present some FP indications, nevertheless, to date, a lot of FP indications exist and the authors should at least mentioned some important indication as endometriosis.
- The first part of the review called “experimental research in fertility preservation ” is not well named because it only deals with pharmacological fertility preservative substances which can be administered prior or during ovarian-toxic treatment. Nevertheless, other experimental research in the fiel of FP are currently in progress, but authors don’t mention it.
Moreover, as this strategy, except for GnRHa, is experimental and test only in animal models for the moment, I think that this part should be at the end of the manuscript instead of the beginning.
Author Response
This is a well written review about fertility preservation for women and girls.
I have just some concerns listed below:
- In vitro maturation (IVM) technique is not mentioned in this review although it “ should be regarded as an innovative FP procedure “ as recommended by ESHRE fertility preservation guideline published in 2020. Even if oocyte or embryo cryopreservation after IVM is less effective than after COS, it can proposed in specific indication or when letrozole cannot be proposed.
I think that this technique should be at least mentioned in this review ( for example in the breast cancer part as a live birth was recently described Grynberg et al, 2020)
Response: Thank you for your positive comments. We have now added IVM as an innovative method for FP in the chapter of breast cancer and provided the suggested reference.
- Authors decided to present some FP indications, nevertheless, to date, a lot of FP indications exist and the authors should at least mentioned some important indication as endometriosis.
Response: We have chosen some hot topics and could not possibly discuss all aspects or indications for fertility preservation for benign diseases. In the last paragraph of the introduction we acknowledge the increasing number of indications, and we have added this text: “The list of indications is growing, ranging from common gynecologic diseases such as endometriosis to chronic autoimmune diseases or rare genetic syndromes, and the number of patients who undergo procedures for fertility preservation is continuously increasing.
The first part of the review called “experimental research in fertility preservation ” is not well named because it only deals with pharmacological fertility preservative substances which can be administered prior or during ovarian-toxic treatment. Nevertheless, other experimental research in the fiel of FP are currently in progress, but authors don’t mention it.
Moreover, as this strategy, except for GnRHa, is experimental and test only in animal models for the moment, I think that this part should be at the end of the manuscript instead of the beginning.
Response: Thank you for your suggestion. We have moved that chapter to the end and changed the title to a more appropriate one, following the reviewer’s comments.
Reviewer 4 Report
This is an updated and comprehensive review on fertility preservation in young women and pre-pubertal girls. Authors are key opinion leaders of the field and give readers a balanced overview of the standard operating procedures used to reduce the risk of infertility after breast and gynecological cancer treatment, but also in patients with Turner syndrome. Mechanisms of gonadotoxicity and gonadoprotection are presented, including recent evidences. A critical reappraisal of the methodological issues of the published clinical data is also added to help understanding the present limitations. Future venues and research developments are highlighted, with a practical approach particularly useful for clinicians.
The paper is well written and clear.
Only few remarks:
- As most of the paper deals with young girls and women, I would remove the parts related to young boys and men in paragraph 3 and their related references.
- There are some typos that should be corrected: page 3, line 119: “femurs”; page 6, line 215: “aortic” dissection; page 8, line 314: “MRI”; page 9, line 367: “New Zealand”
Author Response
Thank you for your constructive comments. The section about children has been revised according to the reviewer’s comment to highlight the difficulties in this specific patient group, and with specific focus on females. We have removed the references related to male patients with one exception, that clearly states that invasive procedures in children should not be planned and performed if the health of the child may be compromised by a surgical procedure. That specific study is needed to support our statement, as it clearly discusses the issues with children, although the patient population encompassed prepubertal and pubertal boys.
We have also corrected the typos. The page numbers have been changed due to the revision but the corrections are highlighted in yellow in the document.
Round 2
Reviewer 2 Report
Thank you for your revisions